# Targeting Neutrophil Extracellular Traps for Stroke Prognosis: A Promising Path

Eirini Liaptsi [1,†], Ermis Merkouris [1,†], Efthymia Polatidou [1,†], Dimitrios Tsiptsios [1,*], Aimilios Gkantzios [1],
Christos Kokkotis [2], Foivos Petridis [3], Foteini Christidi [1], Stella Karatzetzou [1], Christos Karaoglanis [1],
Anna-Maria Tsagkalidi [1], Nikolaos Chouliaras [1], Konstantinos Tsamakis [4], Maria Protopapa [2],
Dimitrios Pantazis-Pergaminelis [2], Panagiotis Skendros [5], Nikolaos Aggelousis [2] and Konstantinos Vadikolias [1]

1   Neurology Department, Democritus University of Thrace, 68100 Alexandroupolis, Greece;
    liaptsi.eirini@hotmail.com (E.L.); ermimerk@med.duth.gr (E.M.); evthpola@med.duth.gr (E.P.);
    aimilios.gk@gmail.com (A.G.); christidi.f.a@gmail.com (F.C.); skaratzetzou@gmail.com (S.K.);
    chrikara65@med.duth.gr (C.K.); annatsag@med.duth.g (A.-M.T.); nikochou4@med.duth.gr (N.C.);
    vadikosm@yahoo.com (K.V.)
2   Department of Physical Education and Sport Science, Democritus University of Thrace,
    69100 Komotini, Greece; chkokkotis@gmail.com (C.K.); mprotopa@phyed.duth.gr (M.P.);
    dpantazi@phyed.duth.gr (D.-P.-P.); nagelous@phyed.duth.gr (N.A.)
3   Third Department of Neurology, Aristotle University of Thessaloniki, 54124 Thessaloniki, Greece;
    fpetridis83@gmail.com
4   King's College London, Institute of Psychiatry, Psychology and Neuroscience, London SE5 8AF, UK;
    ktsamakis@gmail.com
5   First Department of Internal Medicine, Democritus University of Thrace, 68100 Alexandroupolis, Greece;
    pskendro@med.duth.gr
*   Correspondence: tsiptsios.dimitrios@yahoo.gr
†   These authors contributed equally to this work.

**Abstract:** Stroke has become the first cause of functional disability and one of the leading causes of mortality worldwide. Therefore, it is of crucial importance to develop accurate biomarkers to assess stroke risk and prognosis. Emerging evidence suggests that neutrophil extracellular trap (NET) levels may serve as a valuable biomarker to predict stroke occurrence and functional outcome. NETs are known to create a procoagulant state by serving as a scaffold for tissue factor (TF) and platelets inducing thrombosis by activating coagulation pathways and endothelium. A literature search was conducted in two databases (MEDLINE and Scopus) to trace all relevant studies published between 1 January 2016 and 31 December 2022, addressing the potential utility of NETs as a stroke biomarker. Only full-text articles in English were included. The current review includes thirty-three papers. Elevated NET levels in plasma and thrombi seem to be associated with increased mortality and worse functional outcomes in stroke, with all acute ischemic stroke, intracerebral hemorrhage, and subarachnoid hemorrhage included. Additionally, higher NET levels seem to correlate with worse outcomes after recanalization therapies and are more frequently found in strokes of cardioembolic or cryptogenic origin. Additionally, total neutrophil count in plasma seems also to correlate with stroke severity. Overall, NETs may be a promising predictive tool to assess stroke severity, functional outcome, and response to recanalization therapies.

**Keywords:** neutrophil extracellular traps (NETs); acute ischemic stroke; stroke biomarker; intracerebral hemorrhage; subarachnoid hemorrhage; immune thrombosis

## 1. Introduction

Stroke has become one of the leading causes of mortality in the later years of life and remains the first cause of functional disability worldwide [1]. Annually, the number of patients affected by ischemic stroke across the world exceeds nine million people, making it a major medical burden [2]. Emerging recanalization therapies such as recombinant tissue

plasminogen activator (rt-PA) thrombolysis and mechanical thrombectomy are not always successful, and not every patient is eligible for them due to comorbidities and/or timing constraints. Additionally, the use of antiplatelet and anticoagulant therapy as secondary prevention cannot guarantee that ischemic stroke will not recur.

As a result, identifying reliable biomarkers for evaluating the risk of stroke recurrence and functional outcome becomes of major importance. Encouragingly, other mechanisms implicated in stroke pathogenesis such as neuroinflammation and immunothrombosis begin to interest neuroscientists as potential targets for advanced immunotherapies of use in stroke acute treatment and secondary prevention [3]. The notion of immunothrombosis first emerged a decade ago, and the interplay between innate immunity and thrombosis has rapidly become a hotspot [4].

The innate immune system in physiological circumstances acts via immunothrombosis to reduce pathogen spreading by various mechanisms. Leucocyte elevation, with neutrophil predominance, is frequently found in nearly all acute ischemic stroke (AIS) patients independently of stroke etiology, and it may serve as an interesting biomarker. Intriguingly, it has been recently demonstrated that neutrophils play a key role in immune thrombosis pathogenesis. It is well established that neutrophils may trigger coagulation cascades as well as participate in changes in the vascular endothelium via local inflammation mechanisms leading to thrombosis [4].

Firstly, an interplay between activated platelets and neutrophils, in places of atherosclerotic plaque rupture, has been demonstrated in patients suffering an acute myocardial infarction. One of the most potent neutrophil weapons to this end is the formation of thrombogenic neutrophil extracellular traps (NETs) [5]. Activated neutrophils form NETs by releasing decondensed chromatin from their nuclei. Consequently, NETs are composed of decondensed chromatin lined with granular and cytosolic proteins [6]. Levels of NETs are measured by using markers such as cell-free DNA (cfDNA), myeloperoxidase-histone complexes, DNase activity, and circulating citrullinated histone H3 (citH3). NETs expressing neutrophil tissue factor have been shown to play a key prothrombotic role locally in acute coronary syndrome or systemically in antineutrophil cytoplasmic antibody (ANCA)-associated vasculitis and in the immunothrombotic burden of COVID-19 [5,7,8]. These markers have also been used as biomarkers to predict ischemia in peripheral artery disease and coronary disease [9–11]. Recent evidence indicates that neuroinflammation, especially via NET formation, is also implicated in the pathogenesis of arterial thrombosis in cerebral arteries [12]. Moreover, NETs appear to be present in intracranial thrombi by complicating their dissolution and removal, leading to worse clinical outcomes in patients undergoing recanalization therapies [8,13].

Nevertheless, data suggest that systemic inflammation induced by a pathological metabolic state (hyperlipidemia or hyperglycemia) seems to be implicated in atherosclerosis, one of the leading causes of AIS. It is demonstrated that bone marrow hematopoietic cells release leukocytes into the circulation [14], and these leukocyte populations, especially neutrophils via NET release, play a crucial role in the formation of atherosclerotic plaque [15]. Regarding AIS of cardiogenic etiology, other mechanisms are proposed to induce inflammation, such as the activation of the heart endothelium due to hypoxia via hemodynamic changes in atrial fibrillation (AF). These endothelial cells recruit leukocytes by secreting inflammatory factors and chemokines [16].

Recent research shows that NETs create a procoagulant state by serving as a scaffold for red blood cells, platelets, and adhesion molecules inducing thrombosis by activating coagulation pathways [17]. Increased levels of NET markers such as extracellular chromatin and DNA are associated with increased stroke risk in cancer patients [18]. Additionally, new data supporting that NETs may serve as an interesting biomarker for immune-mediated thrombosis have been obtained in the context of the COVID-19 pandemic [8,19]. Overall, it has already been demonstrated that NETs are present in retrieved cerebral thrombi in all etiology of ischemic stroke [18,20].

In addition, current research links elevated NET levels with worse outcomes after intravenous thrombolysis [21]. More precisely, data from animal models and human thrombi analysis show that recanalization time after thrombectomy is affected by the composition of thrombi (red blood cells, fractions of fibrin, and neutrophil extracellular traps). NETs participate in thrombi composition, and their amount seems to increase over time and relate to alterations in fibrin structure [11,17]. Interestingly, stroke etiology seems to affect thrombi size and composition [22,23].

In patients suffering from AIS, neutrophils present a dysregulation in their normal function. It has been demonstrated that neutrophils more frequently form NETs in the AIS setting. Furthermore, even after DNAse-I administration, NETs are more difficult to resolve in AIS patients compared to healthy controls. Interestingly, these findings are apparent even before AIS onset. Consequently, NETs are considered to induce thrombosis rather than just being released as an immune response to the ischemic event. To this end, NETs may be a pharmacological target even for stroke prophylaxis as well [22].

Overall, breakthrough research with emerging evidence establishes that activated neutrophils and NET formation can contribute to the formation of thrombus, leading to ischemic stroke by influencing the physicochemical properties of thrombi in a number of interrelated ways. As a result, NETs may serve as a key biomarker for predicting stroke recurrence and prognosis. A better understanding of the underlying pathophysiological mechanisms may lead to a new future target for acute and preventive stroke therapies. Our narrative review aims to gather evidence from recent studies to elucidate the potential utility of the NET markers measurement to predict stroke severity, functional outcome, mortality risk, and response to recanalization therapies.

## 2. Materials and Methods

The Preferred Reporting Items for Systematic Reviews (PRISMA) checklist (CRD42023451043) was used to guide this study. The methodology used in our study was a priori designed.

### 2.1. Search Strategy

A literature research of two databases (MEDLINE and Science Direct) was conducted by two investigators (EL and EP) to trace all relevant studies published between 1 January 2016 and 31 December 2022 using either "Neutrophil extracellular traps" or "NETs" and "stroke" or "ischemic stroke" as a search criterion. Also, the terms "stroke prognosis" or "stroke outcome" or "stroke severity" were used as a second search criterion. The retrieved articles were also hand-searched for any further potential eligible articles. Any disagreement regarding the screening or selection process was resolved by a third investigator (DT) until a consensus was reached.

### 2.2. Selection Criteria

Only full-text original research articles published in the English language were included. Secondary analyses, reviews, guidelines, meeting summaries, comments, unpublished abstracts, or studies conducted on animals were excluded. There was no restriction on study design or sample characteristics.

### 2.3. Data Extraction

Data extraction was performed using a predefined data form created in Excel 16.54. We recorded the author, year of publication, study design, type of stroke, number of participants, age and gender of participants, timing of sampling, scales utilized for assessing stroke severity, and main results.

### 2.4. Data Analysis

No statistical analysis or meta-analysis was performed due to the high heterogeneity among studies. Thus, data were only descriptively analyzed.

## 3. Results

### 3.1. Database Search

Overall, 438 records were retrieved from the database search. Duplicates and irrelevant studies were excluded. Then, 88 records were screened with our secondary search criteria, and 54 full-text articles were selected as potentially eligible for our review. After screening the full text of the articles, 33 studies were ultimately eligible for inclusion (Figure 1).

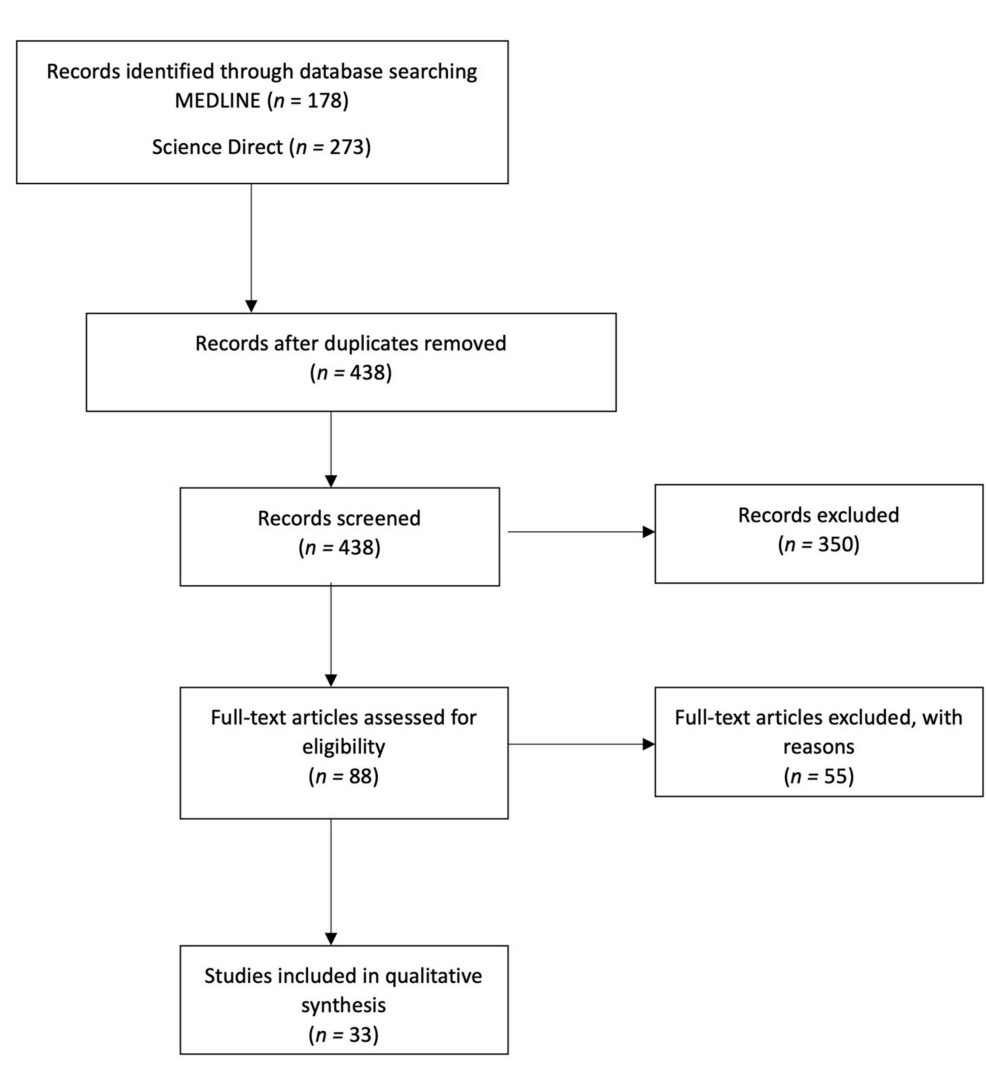

**Figure 1.** Study flow chart (PRISMA diagram).

### 3.2. Study Characteristics

Thirty-three publications fulfilled our inclusion criteria. They were classified into three groups based on the type of sample used to assess NETosis biomarkers. The first group comprised eighteen studies examining NET content from mechanical thrombectomy retrieved thrombi and correlations with stroke outcome and stroke severity. The second group included twenty-one studies measuring neutrophils and NET marker levels in arterial or venous peripheral blood samples (plasma or serum) and examining their association with stroke outcome. Finally, three studies on NET deposition on histopathological specimens of the post-mortem brain tissue of stroke patients were included. The characteristics and main findings of the included studies are presented in Table S1, Supplementary Materials.

### 3.3. Study Design

Most of the studies included in our review were prospective multi-center or mono-center studies, while 3 retrospective and 1 post-mortem study were included.

### 3.4. Stroke Patient Groups

The total number of stroke patients included in all studies ranged from *n* = 7 [11] to *n* = 336 [16]. Across the 33 studies, 25 studies had stroke patient sample sizes between 1 and 100 patients, 5 studies had patient sample sizes between 101 and 200, and 3 studies had patient sample sizes between 201 and 300. In fifteen of the studies presented, there was a control group of healthy individuals.

### 3.5. Demographic and Clinical Profiles

The mean/median patient ages ranged from 43.6. years old [24] to 78 years old [25]. In total, 29 studies examined patients with IS, 2 studies examined patients with subarachnoid hemorrhage (SAH), and 2 studies examined patients with intracerebral hemorrhage (ICH).

### 3.6. Time of Sampling

Thrombus material was collected during endovascular thrombectomy in 17 studies. Peripheral blood samples were collected in 20 of the presented studies. Among them, serum samples were collected upon admission in 12 studies (blood sampling was performed in the first 24 h to 48 h); in 2 of the aforementioned studies, blood sampling was performed upon admission and 3 and 4 days after stroke onset [3,26]; in 1 study, blood sampling was performed 3 months after stroke onset; in 2 studies, blood sampling was performed 48 h after admission; and in another study, blood samples took place after 72 h. In 2 studies, the timing of blood sampling was not measured.

### 3.7. Scales of Stroke Severity and Prognosis/Clinical Outcome

The National Institutes of Health Stroke Scale (NIHSS) score was used in 21 out of the 33 studies to assess stroke severity upon admission, while the modified Rankin Scale (mRS) was used in 11 studies to assess stroke recovery. Additionally, the thrombolysis in cerebral infarction (TICI) score was calculated in 8 studies, while other scores such as the Glasgow Coma Scale (GCS), Trial of Org 10172 in Acute Stroke Treatment (TOAST), and ICH score were rarely used.

### 3.8. Presentation of Main Findings

3.8.1. Abundant NET Formation in AIS Patients' Thrombi and Peripheral Blood

Data presented here elucidated that there was abundant NET formation in AIS patients' thrombi retrieved during mechanical thrombectomy [12,16,26,27]. NETs were present in 100% of thrombi retrieved from AIS patients, whereas this percentage was not that high for thrombi retrieved from patients suffering acute myocardial infarction [28]. Neutrophil and NET marker levels in lysis supernatants from retrieved thrombi for both non-COVID-19 and COVID-19 patients suffering from AIS were equally high regardless of COVID-19 infection status [19]. Moreover, neutrophils in peripheral blood were significantly higher in COVID-19 patients suffering from stroke than in controls, whereas thrombi retrieved during mechanical thrombectomy showed no significant difference in terms of NET composition between the two groups [19]. NET components in AIS thrombi were correlated parabolically with systemic inflammatory markers and positively with patients' age [18]. Additionally, NETs were found to be relatively low in normoglycemic stroke patients and much higher in diabetic and acute hyperglycemic patient subgroups [26].

Furthermore, NET content in thrombi was found to significantly correlate with NET content in peripheral blood [29]. Levels of NET markers and absolute granulocyte count in peripheral blood were significantly higher in patients suffering an acute stroke compared to controls [3,22,30]. Similarly, concentrations of double-stranded DNA (dsDNA) as a NET marker were statistically higher in patients with AIS compared to controls [13]. Increased

plasma DNA levels were associated with cancer-related stroke as well [10]. Higher NET levels in serum were significantly linked to atheroma plaque vulnerability in a subgroup analysis of patients who had never received either statins or antithrombotic treatment [31].

### 3.8.2. Correlation between Elevated Neutrophil Total Count, NET Marker Levels in Plasma, and Ischemic Stroke Severity Functional Outcome

The neutrophil total count in serum as well as NET marker levels in AIS patients' plasma seemed elevated in acute stroke and correlated with stroke severity. The number of serum neutrophils was positively associated with stroke severity as assessed by the NIHSS at admission [3,22,32] and discharge [3,32]. A higher neutrophil fraction was found in patients with poor outcomes at 3 months after stroke onset, while patients with lower neutrophil counts upon admission were linked to less severe NIHSS scores at stroke onset. The neutrophil number was significantly correlated with the number of NETs—they were colocalized, confirming the relationship between neutrophils and NET formation [33]. Neutrophils and neutrophil to lymphocyte ratio (NLR) were demonstrated to be higher in AIS patients with or without infection compared to controls. In AIS patients without infection, NLR was positively correlated with higher NIHSS scores. Additionally, in patients with infarction > 1.5 cm, both neutrophil counts and NLR positively correlated with patients' infarct sizes [34]. Moreover, NETs markers were elevated in plasma of patients with carotid artery occlusion compared to controls [35]. However, in one study including stroke patients presenting with thrombotic thrombocytopenic purpura with small ischemic stroke, they had higher levels of NETs than those with larger strokes [36]. Finally, higher NET levels were associated with a higher risk of all-cause mortality, cardiovascular mortality, composite cardiovascular events, and ischemic stroke [3,37].

### 3.8.3. Link between NETs Presence, the Age of Thrombi, and Success of Recanalization Therapies

NET levels in the thrombus content seemed associated with longer time intervals from stroke onset and worse outcomes after recanalization therapies. Datsi et al. showed that patients with symptom onset longer than 4.5 h had significantly increased NET formation in thrombi. Additionally, sera obtained from AIS patients were proven to significantly promote NETosis in untreated neutrophils despite significantly higher DNase-I levels and poorer clearance of already formed NETs [22]. Abbasi et al. showed that H3cit levels were measured in thrombus fractions of patients undergoing mechanical thrombectomy, and higher H3cit levels were significantly correlated with longer recanalization times. More precisely, older thrombi (>4 h from stroke onset) were richer in H3cit and became stiffer, suggesting that rtPA treatment past 4 h may not be able to adequately dissolve thrombi [23]. Zhang et al. observed a significant increase in NET markers after rt-PA administration in the no-improvement group. Circulating NET markers were positively correlated with plasma levels of procoagulant biomarker indicators such as d-dimers and fibrinogen [38]. Additionally, Kitano et al. demonstrated that older thrombi had higher NET concentrations compared to fresh thrombi. Moreover, more device passes were required to retrieve older thrombi and achieve reperfusion, and this led to worse functional outcomes after mechanical thrombectomy [39]. Novotny et al. confirmed the abovementioned hypothesis, showing that NETs were associated with firmer thrombus composition and worse clinical outcomes, assessed by NIHSS score [20]. To the same end, Ducroux et al. showed that NET content was positively associated with longer endovascular procedures and a higher number of device passes [16]. Laridan et al. also found an abundant presence of NETs in ischemic stroke thrombi, demonstrating that older thrombi (>1 day) contained significantly higher amounts of neutrophils and H3Cit compared to fresh thrombi [32,40].

### 3.8.4. NETs' Correlation with AIS of Cardioembolic or Cryptogenic Origin

Higher levels of neutrophils and/or NETs in AIS patients' peripheral blood and thrombi seem to be correlated with stroke of cardioembolic or cryptogenic origin. Valles et al. indicated that citH3 and cfDNA levels were higher in patients with cardioembolic

stroke compared to other stroke subtypes, suggesting a higher immune-inflammatory activation in cardioembolic stroke [3]. On the same note, Genchi et al. demonstrated that thrombi of cardioembolic origin were significantly richer in NET content compared with thrombi due to large artery atherosclerosis. Interestingly, no difference in total neutrophil count was found, suggesting a different activation state of neutrophils [29]. Similarly, Laridan et al. showed that thrombi of cardioembolic origin contained nearly double the number of NETs compared to non-cardioembolic thrombi [40], and Essig et al. also found that neutrophil numbers were higher in thrombemboli of presumed cardioembolic origin and in thrombemboli of cryptogenic origin compared to non-cardioembolic thrombemboli [32]. Apart from that, in a study conducted by Arroyo et al., all NET markers were also significantly elevated in patients with AF, especially the levels of cfDNA and H3Cit in the cardioembolic stroke subtype. They also demonstrated that the level of neutrophil elastase (NE), a NET marker, was associated with a higher risk of all-cause mortality, cardiovascular mortality, composite cardiovascular events, and ischemic stroke. In addition, the predictive power for the above items was slightly increased after adding NE to the clinical scores, although it was not significantly different from the original CHA2DS2-VASc score [37]. Additionally, Novotny et al. showed that thrombi classified as TOAST-1 presented fewer NETs compared to thrombi classified as TOAST-5, implying that NETs could serve as a potential biomarker, especially in cryptogenic stroke [20]. Cha et al. demonstrated that AF was more frequent in patients with a higher neutrophil count [33]. Moreover, Molek et al. found that increased plasma H3cit concentrations led to an increased thromboembolic profile in AF patients in long-term follow-up [41].

### 3.8.5. NETs Correlate with Worse Functional Outcome in SAH and ICH

Higher NET levels were also found in patients presenting with SAH and ICH and correlated with unfavorable outcomes. Firstly, Zeng et al. showed that the expression of NETs was significantly increased at 12 h in peripheral blood and 24 h in the brain after SAH [21]. Apart from that, Witsch et al. showed that high concentrations of MPO-DNA complexes were present in all patients with aneurysmal SAH on admission regardless of the development of delayed cerebral ischemia (DCI) or clinical vasospasm. Additionally, increased NLR as a marker of post-SAH inflammation was consistently linked in different studies with DCI [42]. Finally, in autopsies of patients who died of spontaneous ICH, Puy et al. found that the total number of neutrophils was increased in the area of the hematoma compared with the contralateral sections in control groups, in which no neutrophils were observed within the tissue. Neutrophil infiltration continued to increase 12–24 h after the incident, which the authors termed the first wave, but they also observed a distinct second wave that occurred between days 8 and 15 after the incident [25].

### 4. Discussion

Overall, as stroke becomes a major threat to public health by being one of the first causes of mortality and the first cause of disability worldwide, there is an urgent need to define stroke biomarkers that may serve to predict stroke severity and prognosis. Emerging data presented here showed that an increased concentration of absolute neutrophil count, NLR, and NET content both in peripheral blood and intracranial thrombi of stroke patients was strongly associated with stroke severity and unfavorable outcomes. Moreover, these markers' concentration seemed to increase as time passed from stroke onset. Even more, high NET levels seemed to be a predictor of inadequate response to recanalization therapies. Interestingly, neutrophils and NETs were presented to be an indirect indicator of stroke etiology, significantly linked with stroke of cardioembolic or cryptogenic origin.

### 4.1. NETs Presence in Stroke Patients' Thrombi, Peripheral Blood, and Brain Tissue: Timing and Pathophysiological Implications

To begin with, it is demonstrated that NETs were present in AIS patients' thrombi regardless of endovascular treatment, and their levels increased over time [12,16,22,27]. An

interesting study by Laridan et al. demonstrated that the extracellular DNA content of AIS thrombi was similar to that of coronary artery disease patients, and this correlated positively with markers of systemic inflammation as well as with patients' age [18]. Intriguingly, older thrombi (>1 day) contained significantly higher amounts of neutrophils compared to fresh thrombi (<1 day) [40]. Additionally, increased DNase-I levels were observed in AIS patients when the time passed from symptom onset > 4.5 h [22], and this finding could be of interest as neutrophil count and NET markers could potentially be tested as an indirect marker to estimate the timing of stroke onset for wake-up strokes and/or AIS with an unknown time of onset. However, dynamic studies demonstrated that neutrophils derived from stroke patients seem to form easier NETs compared to controls, and NETs of stroke patients compared to healthy subjects were more difficult to dissolve after DNAse-I treatment. Therefore, it is speculated that neutrophil dysfunction may occur before stroke onset. As a result, NET formation was considered to play a key role in AIS pathogenesis rather than being solely an immune response to stroke onset [22]. Thus, further studies investigating stroke incidence and its correlation to NETosis markers in a healthy population are required.

It is crucial here to elucidate that NET content in thrombi seemed to correlate with NET content in blood, and NET levels were increased in the peripheral blood of stroke patients [11]. As most stroke patients were not eligible for mechanical thrombectomy and thrombi are not easy to access and analyze in everyday clinical practice, the possibility of measuring NET levels in peripheral blood is of major importance. Many of the aforementioned studies suggested that levels of different NET markers such as MPO-DNA, citH3, cf DNA, nucleosomes, and DNAse-I were shown to be significantly higher after stroke onset in AIS patients' peripheral blood compared to controls [3,13,22,30,38]. Increased peripheral blood DNA was associated with cancer-related stroke as well [10]. Thalin et al. resumed that a procoagulant state and a three-fold increase in the NET-specific marker H3cit were found in the plasma of cancer patients presenting with stroke. The authors proposed that the intense inflammatory response and hypoxic milieu surrounding the infarct could, in this setting, activate neutrophils to produce more NETs, inducing a local NET burden at the location of the infarction, which could subsequently contribute to thrombus growth and stabilization [17]. Intriguingly, a positive correlation between age and citH3 was made in one study [3].

Interestingly, AIS patients presented significantly elevated levels of absolute granulocyte count and elevated neutrophils [13,22,38] and NLR in peripheral blood compared to controls [31]. This finding was independent of infection status [34]. In a post-mortem histopathological study of intracerebral hemorrhage patients, neutrophils were shown to be increased within the hematoma and in tissue surrounding hematoma areas compared to control contralateral sections in which no neutrophils were observed within the tissue, while NET markers were detected in half of the cases. It is worth mentioning that, here also, the presence of both neutrophils and NETs was time-dependent following a two-wave pattern: during the first 72 h and between 8 and 15 days after ICH onset [25]. Undeniably, further studies are required to explore the kinetics of neutrophil gathering and NET formation after stroke onset; this may be of crucial importance to evaluate, whereas neutrophil count and NET levels can accurately predict the timing of stroke onset.

On another note, in SAH patients, MPO-DNA levels seemed to decrease on day 4 in patients with DCI and patients with clinical vasospasm [42]. This finding also supports the abovementioned idea that more research is needed to assess the potential use of NETs as a potent biomarker to predict time of onset. However, high levels of MPO-DNA complexes were detected in all patients with aneurysmal SAH, independently of DCI and/or clinical vasospasm presented at admission [42], and H3cit levels were also higher in patients with aneurysmal SAH compared with healthy controls [21]. Inflammation and oxidative stress are speculated to be vital contributors to the pathological process of brain injury after stroke, and NET release many cytotoxic proteases and therefore induce endothelial cell damage and disrupt vascular homeostasis [36]. The authors hypothesized that the migration of neutrophils into the brain after SAH was a consequence of blood–brain barrier damage.

They suggest that increased NETosis caused inflammation, brain edema, and neuronal damage within the hemorrhage area, while the inhibition of NETs reduced not only the extent of brain injury but also angiogenic edema. Consequently, they proposed DNase I as a potential therapeutic target for the prevention of early brain injury after SAH [22].

### 4.2. Neutrophil Count, NLR, and NETs as Biomarkers of Stroke Severity, Unfavorable Outcomes, and All-Cause Mortality

To start with, in a study conducted by Denorme et al., plasma H3cit and MPO-DNA levels positively correlated with worse clinical outcomes after stroke (using mRS at discharge as an assessment tool). In contrast, stroke severity at admission did not significantly correlate with these immune thrombosis biomarkers. The authors also underlined that co-morbidity, thrombolysis, the use of antiplatelet drugs or anticoagulation drugs, or ischemic stroke etiology did not influence their results [43].

Conflicting data come from Valles et al., as a positive correlation between stroke severity (assessed by NIHSS score) and NET markers both at admission and at discharge was demonstrated [3]. On the same note, neutrophil count was shown to be associated with worse outcomes (measured by using mRS at 3 months) and stroke severity (assessed with NIHSS score) at admission [22,33]. Even more, in AIS patients without infection, NLR was significantly correlated with higher NIHSS scores at admission, while for patients with infarction > 1.5 cm, both neutrophil counts and NLR positively correlated with patients' infarct sizes, as Cai et al. supported [34]. They suggested that NET formation within blood vessels and cerebral parenchyma in the peri-infarct cortical areas may impair revascularization and vascular remodeling after stroke and therefore be associated with higher NIHSS scores [32]. A significant positive correlation between plasma H3cit levels and clinical severity was demonstrated for patients suffering from aneurysmal SAH as well [42]. Another interesting finding in terms of post-stroke mortality risk comes from Cai et al., as patients who died at one-year clinical follow-up from all causes had significantly higher citH3 at stroke onset than survivors [3]. These results were further supported by a study conducted by Arroyo et al., who showed that NE levels as a marker of NETs were significantly associated with a higher risk of all-cause mortality, cardiovascular mortality, composite cardiovascular events, and ischemic stroke [37]. On the other hand, Ducroux et al. failed to prove any significant correlation between NET content in thrombi and stroke pathogenesis, 3-month functional outcome, or final thrombolysis in cerebral infarction (TICI) score [16].

Overall, a paucity of data and the high heterogenicity of findings across studies make it difficult to safely assess the potential to use high neutrophils and NET levels to predict worse clinical outcomes and stroke severity. However, encouraging data from several studies seem to support the hypothesis that high levels of NETosis markers may serve as a biomarker, especially for worse clinical outcomes and higher mortality risk after stroke, even though extensive research is absolutely required.

### 4.3. NETs as a Prognostic Tool for Predicting Response to Recanalization Therapies

Nevertheless, NET levels seem to be an interesting predictive tool to predict recanalization therapies' success and functional outcomes. NET formation has been associated with possible resistance to rt-PA administration. Interestingly, published data indicated that when DNAse I was administered ex vivo to inhibit NET formation, the thrombolysis process seemed to be accelerated. These results suggested that NETs could be a potential pharmacological target to optimize the efficiency of intravenous thrombolysis [3,22,28], as NETs presence in thrombi was linked to firmer thrombus composition and malign stroke outcomes in several studies [14,22]. Several groups' preliminary data implied that DNAse 1 administration ex vivo improved thrombolysis outcomes probably by potentiating fibrinolysis [40]. For instance, Ducroux et al. proved that co-administering rtPA and DNAse 1 accelerated ex vivo thrombolysis when compared with rtPA or DNAse 1 administration alone [16]. Zeng et al. also showed that an injection of DNase I significantly

reduced neurological damage when compared with injecting saline, and this was a result of pharmacologically inhibiting NET formation [21].

On the other hand, Cha et al. failed to show any significant correlation between neutrophil count and either recanalization after thrombectomy (using TICI score as an assessment tool) or onset to recanalization time [33], but Ducroux et al. demonstrated that NET content in thrombi was associated with endovascular procedure length and device number of passes [16]. Zeng et al. also proved that H3Cit thrombus fractions were significantly higher in the delayed recanalization time group [21], whereas Kitano et al. presented that older thrombi had a greater extent of NETosis and that more device passes before reperfusion were necessary for older thrombi and were associated with poorer functional outcomes [39]. Moreover, data from a study conducted by Orban et al. elucidated that after thrombolysis, NET markers significantly increased in the no-improvement group. The authors speculated that the pathophysiological implications of the NET interaction with fibrin could result in the formation of denser thrombi that are more resistant and more difficult to dissolve [9].

A retrospective study analyzing thrombi composition retrieved from patients who underwent thrombectomy showed that thrombi were richer in fibrin and NET markers, such as citH3, when the time for recanalization was longer, and this result was suggested by the authors to be due to coagulation pathways being activated as time passes. This difference in thrombus content would make thrombi less viscoelastic and thus mechanical thrombectomy more difficult. It is noteworthy that the contraction of thrombus, which has been shown to occur over time after stroke onset, could favor the reduction in vessel occlusion and improve cerebral blood flow due to the changes in thrombus content but also led to stiffer thrombi that are more resistant to external fibrinolysis. This finding could possibly explain why intravenous thrombolysis is ineffective after four hours [23].

### 4.4. NETs as Predictor of Stroke Risk

Interesting data in terms of using NETs as a biomarker to predict stroke occurrence risk come from Zhang et al., who showed that NET marker levels in plasma were higher in patients with symptomatic carotid stenosis compared with healthy controls and asymptomatic patients [24]. On the same note, higher levels of NETs were significantly associated with more vulnerable atherosclerotic plaques in a subgroup of patients naïve to hypolipidemic and antithrombotic medication in a study conducted by de Vries et al. [31]. Nevertheless, further prospective studies in healthy populations are needed to support if this finding exists prior to stroke onset in patients with carotid artery disease and to thus evaluate if elevated NET levels may serve as a predictive tool. The researchers suggested that statins and antithrombotic treatment could interfere with a variety of pathophysiological mechanisms implicated in atherosclerotic plaque formation and vulnerability, such as endothelial dysfunction, local inflammation, and immune thrombosis. The underlying mechanism proposed is that as inflammatory cells, especially neutrophils, present higher rates of oxygen consumption due to interplaque hemorrhage, they may trigger neovascularization and thus potentially increase hemorrhage [31]. Of note, interesting data from recent studies linked increased plasma NET levels in patients with stroke risk factors, such as atrial fibrillation [41], dyslipidemia [44] and essential hypertension [45].

Intriguingly, Molek et al. found that increased plasma H3cit concentrations led to an increased thromboembolic profile in AF patients in long-term follow-up [15]. They observed that higher concentrations of this specific marker were correlated with denser fibrin clots composed of thin fibers that are more resistant to thrombolysis, suggesting that enhanced NETosis from neutrophils may play a role in the prothrombotic properties of fibrin clots in thrombi from AF patients. Furthermore, they proposed NET marker citH3 as an independent predictor of ischemic stroke or transient ischemic attack in patients with a history of atrial fibrillation on anticoagulation treatment [15]. Finally, NETs presence in blood, as well as increased NLR was associated with carotid plaque vulnerability and in-

creased stroke risk for patients presented with carotid artery stenosis, in a study conducted by Shimonaga et al. [46].

*4.5. NET Levels as an Indirect Marker of Stroke Etiology*

Another finding worth mentioning in our review was the fact that NETs could potentially serve as a tool to indirectly predict stroke etiology and thus facilitate secondary prevention. More precisely, higher neutrophil count was shown to be significantly associated with stroke due to atrial fibrillation (AF) [33], while NET marker cfDNA was demonstrated to be significantly higher in the plasma of AF patients compared to controls [33]. Valles et al. also showed that markers of NETs were significantly elevated in patients with a history of AF, and levels of cfDNA and CitH3 were especially higher in patients with cardioembolic stroke [3]. Moreover, thrombi of cardioembolic origin contained nearly double the number of NETs compared to non-cardioembolic thrombi in a study conducted by Laridan et al., and even Genchi et al. also showed that NET content was significantly increased in cardioembolic compared to large artery atherosclerosis thrombi [11]. Novotny et al. also demonstrated that higher levels of NETs were found in thrombi of patients suffering from AIS of cardioembolic and cryptogenic origin [20]. Many studies speculated that NET dysregulation may trigger specific pathophysiological mechanisms contributing to thrombotic complications of unknown cause, and this may be of interest for cryptogenic embolic strokes, which account for up to 30% of all AIS [15].

## 5. Limitations and Future Directions

Our systematic review has several limitations. First, we cannot completely rule out the possibility that the included studies published by the same research team used data from the same patient groups. Secondly, the studies presented in this review may succumb to selection bias as researchers often included patients with mild to moderate strokes because patients with severe disability, patients unable to undergo MRI scans, and patients with severe comorbidities who could not give consent were sometimes excluded. Moreover, other studies included in our review concerned specific populations of patients (COVID-19 patients, TTP patients, and cancer patients), limiting the applicability of the abovementioned findings to every patient affected by stroke. Furthermore, in most studies, no stratification of patients based on sociodemographic characteristics was made; so, potentially, important confounders have not been taken under consideration. However, we conducted here an in-depth evaluation of patients' sociodemographic and clinical variables, which may provide an indirect estimate of their role in acute stroke and inform the design of future studies.

As it is made clear from all of the above, studies included here present some important biases. Most of the studies were single-centered, and they reported data analyzing only a small number of patients. Moreover, even though most of them were prospective, there were a few retrospective studies presented, a fact that affects the accuracy of the data collected. Additionally, as thrombi retrieved for mechanical thrombectomy were analyzed to assess NET levels in most studies, we should interpret these data with caution, as technical issues such as the difficulty of removing the thrombus en bloc and device manipulations during the procedure leading to thrombus fragmentation may result in different measurements due to endothelial activation. Furthermore, some studies measured NETs only in serum and not in plasma, potentially leading to an overestimation of NETs presence, as serum preparation could enhance the release of NETs during the coagulation phase. Finally, different NET markers were examined in each study, potentially leading to highly heterogeneous results.

As far as the studied population is concerned, there was high heterogenicity among patients, as some studies did not have a control group, and many of the studies provided little or no information regarding patient treatments and other possible confounders such as stroke etiology, stroke subtype, and AIS location. Lastly, unfortunately, no long-term

clinical outcome was reported in the majority of studies, and only a few assessed stroke clinical severity at admission.

To sum up, many research groups in recent years have tried to target neuroinflammation and immune thrombosis mechanisms to optimize stroke treatment. However, in a previous review published by Drieu et al., the use of anti-inflammatory agents to this end has been discussed, and no evidence of beneficial effects has been shown. Targeting neutrophils in stroke by using different monoclonal antibodies has been demonstrated not to correlate with improved functional outcomes [47]. Nevertheless, the most discussed trial on the subject was published 20 years ago, using a selective antagonist of the CD11b integrin of MAC-1 (CD11b/CD18) to inhibit the migration and infiltration of neutrophils within a 6 h window from stroke symptom onset, and it was stopped due to futility [48]. In recent years, to the best of our knowledge, there is little published evidence investigating the utility of anti-inflammatory medications to prevent major cardiovascular events after stroke onset [49]. However, a recent trial by Tardif et al. presented some promising findings, demonstrating that low-dose colchicine administration after acute myocardial infarction reduced cardiovascular events and was linked to lower mortality rates. Intriguingly, the authors commented that recruited patients receiving colchicine were less likely to present with a stroke or to be urgently hospitalized for angina [50]. Thus, it seems to be of great interest to further study anti-inflammatory medication in acute stroke settings or even for secondary prevention.

On the same note, recent guidelines of the European Alliance of Associations of Rheumatology (EULAR) recommend the administration of hydroxychloroquine (HCQ)—a common antirheumatic drug—in patients with systemic lupus erythematosus and antiphospholipid syndrome to reduce cardiovascular risk. Interestingly, HCQ inhibits autophagy, and several human studies have linked autophagy with NET generation and activity. Undeniably, future randomized clinical trials are warranted to test this hypothesis [51].

Intriguingly though, data from a recent study published by Mitsios et al. illustrate that antithrombotic treatments such as Ticagrelor may play an anti-inflammatory role by attenuating NETosis in myocardial infraction and thus contribute to reducing inflammation and immune-mediated thrombosis [52]. Moreover, in a phase II trial, targeting the C3 complement in severe COVID-19 cases restrained NET release without increasing the risk of secondary infections [53].

As neuroimmunology and neuroinflammation have become golden targets for the treatment of several neurological diseases in the later years of life, such as multiple sclerosis or dementia, there is accumulating evidence that immune mechanisms could be of interest to vascular neurology as well. Therapeutic approaches targeting inflammation-induced thrombosis rather than trying directly to inhibit coagulation mechanisms may serve as a potent alternative to conventional treatments. Anti-inflammatory therapies could even act as valuable adjunctive therapies enhancing benefits and reducing the harm of potential hemorrhagic complications. Conclusively, randomized controlled trials are needed to adequately test whether the use of anti-inflammatory drugs and/or NET inhibitors such as DNAse I may ameliorate stroke outcomes if administered early after stroke onset.

## 6. Conclusions

Overall, total neutrophil count and NET marker levels measured in peripheral blood and in thrombi may serve as promising biomarkers for stroke prognosis. Presented data from recent, mostly prospective, studies elucidated that there was an abundant NET formation in AIS patients' thrombi retrieved during mechanical thrombectomy and in the plasma of stroke patients. Neutrophil total count as well as NET marker levels seemed to be elevated in acute ischemic stroke, SAH, and ICH patients and correlated with stroke severity, unfavorable outcomes, and higher mortality rates. Moreover, higher levels of NETs may serve as a marker of a longer duration from stroke onset and predict worse functional outcomes of recanalization therapies. Interestingly, NET levels may also become a helpful

indicator of AIS etiology, as higher levels of neutrophils and/or NETs in AIS patients' blood and thrombi seemed to be correlated with stroke of cardioembolic or cryptogenic origin.

**Supplementary Materials:** The following supporting information can be downloaded at: https://www.mdpi.com/article/10.3390/neurolint15040076/s1. Table S1: Basic characteristics of the 33 included studies.

**Author Contributions:** Conceptualization, E.L., D.T. and F.C.; methodology, C.K. (Christos Karaoglanis), K.V. and N.A.; validation, E.M.; formal analysis, F.P.; writing—original draft preparation, D.P.-P., E.P., S.K., A.G. and A.-M.T.; writing—review and editing, F.C., D.T., E.L., P.S., K.T. and M.P.; visualization, C.K. (Christos Kokkotis) and N.C. All authors have read and agreed to the published version of the manuscript.

**Funding:** This work was supported by the project "Study of the interrelationships between neuroimaging, neurophysiological and biomechanical biomarkers in stroke rehabilitation (NEURO-BIO-MECH in stroke rehab)" (MIS 5047286), which is implemented under the action of "Support for Regional Excellence" funded by the operational program "Competitiveness, Entrepreneurship and Innovation" (NSRFm2014-2020) and co-financed by Greece and the European Union (the European Regional Development Fund).

**Institutional Review Board Statement:** Not applicable.

**Informed Consent Statement:** Not applicable.

**Data Availability Statement:** All data discussed within this manuscript are available on PubMed.

**Conflicts of Interest:** The authors declare no conflict of interest.

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
