# Peer review of "Targeting Neutrophil Extracellular Traps for Stroke Prognosis: A Promising Path"

_2035-8377, doi:10.3390/neurolint15040076_

Round 1

Reviewer 1 Report

The review article on neutrophil extracellular traps (NET) is timing and interesting. However the authors confuse systemic review with narrative review. The current writing style is systemic review, but the article doesnot include any statistic analysis or meta-analysis because of "high heterogeneity among studies". I suggest rewrite the article in a narative review and resubmit.

In the resubmission, the authors should make clear how NET becomes "biomarkers to assess stroke risk and prognosis" (as indicated in abstract)? How are sensitivity and specificity of NET for stroke patients? 

The Table 1 is long and should be in supplementary document.

There are a few of typos and the manuscript requests a proof-read before future submission.

Author Response

Dear Reviewer,

Many thanks for your prompt response and time spent reviewing our manuscript.

According to your suggestions, appropriate modifications were made to the text as follows:

Comment 1:

The review article on neutrophil extracellular traps (NET) is timing and interesting. However the authors confuse systemic review with narrative review. The current writing style is systemic review, but the article does not include any statistic analysis or meta-analysis because of "high heterogeneity among studies". I suggest rewrite the article in a narative review and resubmit.

According to your suggestion the article is resubmitted as narrative review

Comment 2:

In the resubmission, the authors should make clear how NET becomes "biomarkers to assess stroke risk and prognosis" (as indicated in abstract)? How are sensitivity and specificity of NET for stroke patients? 

A part of our discussion was re-written trying to more coherently answer to the question whereas NETs may serve as a biomarker for stroke risk and prognosis. Gathering data from the 33 studies presented in our review, higher neutrophil absolute count/ neutrophil to lymphocyte ratio and/or along with NETs levels are suggested to predict worse functional outcome at 3 months, be linked with more severe stroke at admission and correlate as with higher mortality rates in most studies. However, conflicting data exist needing further research. In terms of using NETs as a biomarker to assess stroke risk some data suggest that higher NETs levels in patients presented with atherosclerotic disease may be of higher risk of plaque vulnerability/ becoming symptomatic however these data come from only two studies and no prior healthy population has been examined. Unfortunately, no precise data on sensitivity and specificity of NETs were reported beside few studies mentioning that NETs association with stroke severity seem to be independent of infection status.  These remarks summarized above were added to our discussion and highlighted in green.

Comment 3:

The Table 1 is long and should be in supplementary document.

The Table 1 has been moved to the end of the text as a supplementary material as you suggested.

Comments on the Quality of English Language

There are a few of typos and the manuscript requests a proof-read before future submission.

Proof-reading was made.

Looking forward to your follow-up comments.

Yours Sincerely,

Dr Tsiptsios

Reviewer 2 Report

This paper reports on a systematic review of neutrophil extracellular traps for stroke prognosis. The authors concluded that NETs may be a promising 28 predictive tool whose clinical use may significantly improve individualized stroke treatment in both acute settings and secondary prevention.

There are number of issues the authors may wish to address:
1. Introduction - it is too long – much can be moved to Discussion; there is no Study Aim mentioned
2. No mention of biases of each study
3. Results - these tables can be moved to the end of the paper. The Results section should crystalise the key findings from the many papers
4. Discussion – start with a qualitative summary of the main findings. Then explore each main finding in separate paragraphs with supporting literature
5. Overall – the authors should rework the paper and place the appropriate text under the appropriate sections eg result in Results, discussion in Discussion
6. Concern – the authors have not quite answered if  NETs allow prognostication – I see a bit in severity, not much on outcome  except in recanalised patients- how about the vast majority who do not get this therapy? The authors need to place the information in a clinical context that helps the clinician in managing and advising patients on prognosis. The presence of NETs within thrombi or in cardioembolic/cryptogenic strokes is interesting but more of academic than clinical value
7. References – I can’t understand why the first reference is [38] – the refs do not match the text, a re-write is needed
8. There are grammar and vocabulary issues that need attention

There are grammar and vocabulary issues that need attention

Author Response

Dear Reviewer,

Many thanks for your prompt response and time spent reviewing our manuscript.

According to your suggestions, appropriate modifications were made to the text as follows:

Comment 1

Introduction - it is too long – much can be moved to Discussion; there is no Study Aim mentioned

Parts of the introduction were moved in the discussion as suggested,  Study Aim was moved to the final part of the introduction. All aforementioned changes were highlighted in yellow.

Comment 2

No mention of biases of each study

Bias of studies presented were further discussed in the limitations section of the discussion highlighted in yellow.

Comment 3

Results - These tables can be moved to the end of the paper. The Results section should crystalise the key findings from the many papers

Key findings of the studies presented were summarized as proposed in Results section 3.8.  and highlighted in yellow.

Comment 4

 Discussion – start with a qualitative summary of the main findings. Then explore each main finding in separate paragraphs with supporting literature

Discussion has been structured as proposed, qualitative summary of findings has been made in the first paragraph of the discussion. All changes are highlighted in yellow.

Comment 5

Overall – the authors should rework the paper and place the appropriate text under the appropriate sections eg result in Results, discussion in Discussion

Parts of the previous discussion were moved in Results and reformulated and all Discussion has been re-written. All changes highlighted in yellow.

Comment 6

Concern – the authors have not quite answered if  NETs allow prognostication – I see a bit in severity, not much on outcome  except in recanalised patients- how about the vast majority who do not get this therapy? The authors need to place the information in a clinical context that helps the clinician in managing and advising patients on prognosis. The presence of NETs within thrombi or in cardioembolic/cryptogenic strokes is interesting but more of academic than clinical value

Two paragraphs were added in discussion, addressing the question whether Neutrophils and NETs may predict stroke outcome/all-causes mortality and whether/ how they correlate to stroke severity as well as if they may be of use to predict stroke risk in general population. (Changes under the headings : “Neutrophil count, NLR and NETs as a biomarker of stroke severity, unfavorable outcome and all-cause mortality”, “NETs as predictor of stroke risk”.  All changes highlighted in yellow.)

Comment 7. References – I can’t understand why the first reference is [38] – the refs do not match the text, a re-write is needed

References have been re-edited to match to the text.

Comment 8. There are grammar and vocabulary issues that need attention

A proof-reading was made trying to address these issues.

Looking forward to your follow-up comments.

Yours Sincerely,

Dr Tsiptsios

Reviewer 3 Report

Dear Editor,

I reviewed the manuscript detailed below.

“Shall we target Neutrophil Extracellular Traps for Stroke Prognosis?”

The authors investigated in 33 publications the relevance of neutrophil extracellular traps for the prognosis after stroke. I agree with the authors that such an instrument is important, however, I have some points in which conclusions are exaggerated (in my opinion). The manuscript is well written, and the analysis is comprehensive. I am wondering why so many authors are listed.

However, this is manuscript contains interesting findings, reason why I would recommend publishing it.

The following I would recommend to amend:

1. In the abstract I would add a sentence explaining the background of neutrophil extracellular traps.

2. The last sentence in the abstract is speculative. Findings reported in the manuscript do not lead to a more individualized treatment. I would temper this statement.

3. The introduction and the description of the methods is good!

4. The first chapter of the discussion is wrong. Why enumerating propaedeutic things rather than concisely present the main findings. I would reformulate.

5. Throughout the discussion, the word furthermore is frequently used. It leads to the impression, in some parts of the discussion, only reporting rather than discussing was undertaken.

6. Have the authors some explanation regarding the detected findings and the results obtained in the ESUS studies? Some comments on that in chapter 4.4 would be useful (optional).  

7. After incorporating the reported results, could a therapy with anti-inflammatory drugs (eg diclofenac) be of benefit. Would the authors speculate on that?

Dear Editor,

I reviewed the manuscript detailed below.

“Shall we target Neutrophil Extracellular Traps for Stroke Prognosis?”

The authors investigated in 33 publications the relevance of neutrophil extracellular traps for the prognosis after stroke. I agree with the authors that such an instrument is important, however, I have some points in which conclusions are exaggerated (in my opinion). The manuscript is well written, and the analysis is comprehensive. I am wondering why so many authors are listed.

However, this is manuscript contains interesting findings, reason why I would recommend publishing it.

The following I would recommend to amend:

1. In the abstract I would add a sentence explaining the background of neutrophil extracellular traps.

2. The last sentence in the abstract is speculative. Findings reported in the manuscript do not lead to a more individualized treatment. I would temper this statement.

3. The introduction and the description of the methods is good!

4. The first chapter of the discussion is wrong. Why enumerating propaedeutic things rather than concisely present the main findings. I would reformulate.

5. Throughout the discussion, the word furthermore is frequently used. It leads to the impression, in some parts of the discussion, only reporting rather than discussing was undertaken.

6. Have the authors some explanation regarding the detected findings and the results obtained in the ESUS studies? Some comments on that in chapter 4.4 would be useful (optional).  

7. After incorporating the reported results, could a therapy with anti-inflammatory drugs (eg diclofenac) be of benefit. Would the authors speculate on that?

Author Response

Dear Reviewer,

Many thanks for your prompt response and time spent reviewing our manuscript.

Appropriate modifications were made to the text as follows:

Comment 1

In the abstract I would add a sentence explaining the background of neutrophil extracellular traps.

A sentence explaining NETs inducing a procoagulant state and potentializing thrombosis was added-highlighted in blue.

Comment 2

The last sentence in the abstract is speculative. Findings reported in the manuscript do not lead to a more individualized treatment. I would temper this statement.

Last sentence was changed- -highlighted in blue.

Comment 3

The introduction and the description of the methods is good!

Thank you for your kind feedback. Some changes in both introduction and methods were made (highlighted in yellow).

  1. The first chapter of the discussion is wrong. Why enumerating propaedeutic things rather than concisely present the main findings. I would reformulate.

All discussion has been re-written and main findings are presented in the first paragraph, highlighted in yellow.

  1. Throughout the discussion, the word furthermore is frequently used. It leads to the impression, in some parts of the discussion, only reporting rather than discussing was undertaken.

All discussion has been re-written- we tried more this time to discuss findings rather than just list them.

  1. Have the authors some explanation regarding the detected findings and the results obtained in the ESUS studies? Some comments on that in chapter 4.4 would be useful (optional).  

Some comments to this subject have been added to the discussion, highlighted in blue.

  1. After incorporating the reported results, could a therapy with anti-inflammatory drugs (eg diclofenac) be of benefit. Would the authors speculate on that?

A paragraph regarding potential therapeutic targets and use of anti-inflammatory drugs has been added to the discussion, highlighted in blue.

Looking forward to your follow up comments.

Yours Sincerely,

Dr Tsiptsios

Round 2

Reviewer 1 Report

The manuscript needs further improvements as follows:

1. The title would better be a statement rather a question. In addition, will neutrophil extracellular traps be in blood or in thrombi, or both?

2. "Future directions" will better be after "Limitations". You can put them together under a title "Limitations and Future directions"

3. Add the numbers for studies in the Suppl. Table 1, which appear in the References

4. Give numbers (e.g. 4.1, 4.2, ...) for each sections in discussion, as you do in other sections.

5. English expression is hard to follow in a couple of sentences:

e.g. lines 188-190; Lines 206-207; Lines 248-253; Lines 330-335

can you remove sentences 390-392?

6. English tense is not consistent

The manuscript uses three different tense when cite other's studies:

For example

i. authors show + a sentence with present tense

ii. authors showed + a sentence with present tense

iii. authors showed + a sentence with past tense

7. plasma or serum

these two term are randomly used without specifically explaining

8. the use of abbreivations is chaotic

some are not explained at their first appearance, e.g NIHSS (line 121), some donot use in the first apprence but used later appearences, e.g. atrial fibrillation (line 11), some are repeatedly used both abbraviation + full name, e.g NLR (line 358, line 391).

Please thoroughly check out!!

The manuscript needs proofreading.

Author Response

Dear Reviewer,

Many thanks for your additional time spent reviewing our manuscript.

According to your suggestions appropriate modifications were made as follows:

The manuscript needs further improvements as follows:

  1. The title would better be a statement rather a question. In addition, will neutrophil extracellular traps be in blood or in thrombi, or both?

The title has been reformulated. Neutrophil extracellular traps content in both thrombi and blood may serve as a potential biomarker. However, we did not mention this to the title as we use the broad term stroke for both ischemic and hemorrhagic stroke. Measuring NETs in thrombi applies only in a subgroup of patients with ischemic stroke undergoing thrombectomy. If this addition is considered necessary, we are happy to make further changes.

  1. "Future directions" will better be after "Limitations". You can put them together under a title "Limitations and Future directions"

The “future directions” paragraph has been moved under limitations section and the title has been changed as indicated.

  1. Add the numbers for studies in the Suppl. Table 1, which appear in the References

References have been added as indicated.

  1. Give numbers (e.g. 4.1, 4.2, ...) for each sections in discussion, as you do in other sections.

Changes have been made as indicated.

  1. English expression is hard to follow in a couple of sentences:

e.g. lines 188-190; Lines 206-207; Lines 248-253; Lines 330-335

can you remove sentences 390-392?

Lines 188-190: content has been reformulated, Lines 206-207: content has been completely removed, Lines 248-253: content has been reformulated, Lines 330-335: content has been reformulated, Sentences 390-192 have been removed as indicated

  1. English tense is not consistent

The manuscript uses three different tense when cite other's studies:

For example

  1. authors show + a sentence with present tense
  2. authors showed + a sentence with present tense

iii. authors showed + a sentence with past tense

Changes in tense have been made. Present tense was used only in the introduction as general principles of pathophysiology and immunological mechanisms are presented. Past tense was used in methods/results and discussion parts.

  1. plasma or serum

these two term are randomly used without specifically explaining

There is a high heterogenicity among studies on whether serum or plasma was analysed. This issue was addressed in limitations, highlighted in purple.

  1. the use of abbreviations is chaotic

some are not explained at their first appearance, e.g NIHSS (line 121), some donot use in the first apprence but used later appearences, e.g. atrial fibrillation (line 11), some are repeatedly used both abbraviation + full name, e.g NLR (line 358, line 391).

Please thoroughly check out!!

Changes in abbreviations have been made as indicated.

Also some references have been added and some changes have been made in discussion and introduction -highlighted in yellow.

Looking forward to your follow up comments.

Yours Sincerely,

Dr Tsiptsios

Reviewer 2 Report

The authors have adequately addressed my concerns

Author Response

Dear Reviewer,

Many thanks for your prompt response and your effort.

Yours Sincerely,

Dr Tsiptsios